# ‘Toxic Masculinity’: What Is Known about the Role of Androgen Receptors in Head and Neck Squamous Cell Carcinoma

**DOI:** 10.3390/ijms24043766

**Published:** 2023-02-13

**Authors:** Josipa Čonkaš, Maja Sabol, Petar Ozretić

**Affiliations:** Laboratory for Hereditary Cancer, Division of Molecular Medicine, Ruđer Bošković Institute, HR-10000 Zagreb, Croatia

**Keywords:** head and neck squamous cell carcinoma, HNSCC, androgen receptor, AR, membrane androgen receptors, CaV1.2, OXER1, TRPM8

## Abstract

Head and neck squamous cell carcinoma (HNSCC), the most prevalent cancer in the head and neck region, develops from the mucosal epithelium of the upper aerodigestive tract. Its development directly correlates with alcohol and/or tobacco consumption and infection with human papillomavirus. Interestingly, the relative risk for HNSCC is up to five times higher in males, so it is considered that the endocrine microenvironment is another risk factor. A gender-specific risk for HNSCC suggests either the existence of specific risk factors that affect only males or that females have defensive hormonal and metabolic features. In this review, we summarized the current knowledge about the role of both nuclear and membrane androgen receptors (nAR and mARs, respectively) in HNSCC. As expected, the significance of nAR is much better known; it was shown that increased nAR expression was observed in HNSCC, while treatment with dihydrotestosterone increased proliferation, migration, and invasion of HNSCC cells. For only three out of five currently known mARs—TRPM8, CaV1.2, and OXER1—it was shown either their increased expression in various types of HNSCC or that their increased activity enhanced the migration and invasion of HNSCC cells. The primary treatments for HNSCC are surgery and radiotherapy, but targeted immunotherapies are on the rise. On the other hand, given the evidence of elevated nAR expression in HNSCC, this receptor represents a potential target for antiandrogen therapy. Moreover, there is still plenty of room for further examination of mARs’ role in HNSCC diagnosis, prognosis, and treatment.

## 1. Introduction

### 1.1. Head and Neck Squamous Cell Carcinoma

Head and neck squamous cell carcinoma (HNSCC) is the sixth most common cancer worldwide and the most prevalent type of cancer in the head and neck region. In 2020, 879,000 new cases and 445,000 deaths were reported worldwide [1]. The mean age at the diagnosis is 63 years [2] with a five-year survival rate below 50% in the advanced stages of the disease [3]. HNSCC develops from the mucosal epithelial cells of the oral cavity, larynx, nasopharynx, oropharynx, hypopharynx, and sinonasal tract, and its development directly correlates with several risk factors that include alcohol [4] and/or tobacco consumption [5], long-term exposure to environmental pollutants [6], and infection with oncogenic strains of human papillomavirus (HPV) [7] or Epstein–Barr virus (EBV) [8]. Most of the risk factors listed above have a cultural and geographical prevalence due to the different lifestyles. For instance, India’s highest prevalence of oral cavity cancer is connected to tobacco and betel nut chewing and exposure to air pollutants [9].

Furthermore, HPV-positive (HPV+) patients have the highest risk of developing tonsillar (TSCC) and oropharyngeal squamous cell carcinomas (OPSCC), in which the HPV-16 subtype is found in almost 90% of OPSCCs followed by HPV-18 (3%) [10]. On the other hand, laryngeal and oral cavity carcinomas are called HPV-negative (HPV-) because they are associated with smoking and other non-HPV risk factors. As with many other tumors, the formation and progression of HNSCC from the premalignant lesion consists of several steps that include hyperplasia, dysplasia, carcinoma in situ, and invasive carcinoma [11].

After a diagnosis, potential treatments for HNSCC are surgical resection followed by radiotherapy and/or chemotherapy [10]. A lack of screening strategies and consequent disease detection at advanced stages as well as the ineffectiveness of available therapies and treatments leads to a poor prognosis. So far, several molecular biomarkers for clinical diagnosis or prognosis of HNSCC have been approved by the United States Food and Drug Administration (FDA), while only a few of them have proven significant in clinical trials (such as CD44, CD133, and aldehyde dehydrogenase 1 (ALDH1)) [12]. Accordingly, new targeted therapies and predictive, prognostic, and diagnostic biomarkers for the early clinical detection of HNSCC are necessary. Various new biomarkers were evaluated in clinical trials in the last decade; these included hormonal receptors [12,13,14,15].

Interestingly, the relative risk for HNSCC development followed by death is up to five times higher in males compared to females [16]. The endocrine microenvironment is considered to be another risk factor in HNSCC tumorigenesis, although the role of sex hormones in this tumor type is still controversial. However, the gender-specific risk for HNSCC development suggests either the existence of specific risk factors that affect only males or that females have defensive hormonal and metabolic features as a response to common risk factors [17] (Figure 1). In this short but comprehensive review, we present the current knowledge about the role of androgen receptors (ARs) in HNSCC; to the best of our knowledge, this was the first attempt to synthesize what is known about both the nuclear and membrane types of ARs.

### 1.2. Sex Steroid Hormones and Their Receptors

Many types of tumors depend on steroids for growth, proliferation, and survival. Although steroid hormones control development, reproduction, and metabolism as endocrine molecules, they can also mediate tumor initiation and progression. The well-known hormone-sensitive cancers are breast [18], ovarian [19], prostate [20], thyroid [21], testicular [22], and uterine or endometrial cancer [23], whose progression correlates directly with sex steroid hormones [24]. Steroid hormones are derived from cholesterol, and their synthesis is localized in the mitochondria and smooth endoplasmic reticulum of the adrenal cortex, gonads, and placenta [25]. The synthesis of sex hormones is under the control of the hypothalamic–pituitary–gonadal axis. Secretion of gonadotropin-releasing hormone (GnRH) from the hypothalamus activates the anterior pituitary production of follicle-stimulating hormone (FSH) and luteinizing hormone (LH), which control gametogenesis and steroidogenesis [26]. Cholesterol-derived steroid hormones are divided into two main subgroups: sex steroids (androgens, estrogens, and progestogens) and corticosteroids (glucocorticoids and mineralocorticoids) [27]. The main role of steroid hormones is the regulation of gene expression in the cell nucleus or mediation of the rapid modulation of intracellular molecular pathways. Steroids interact with specific nuclear or membrane receptors to accomplish their role [28]. Membrane steroid receptors are integrated transmembrane proteins activated by steroid binding and therefore can activate or inhibit other proteins and modulate different signaling pathways [29]. On the other hand, nuclear hormone receptors function as transcription factors in the nucleus and receptors on the cell membrane. After post-translational covalent attachment of palmitic acid or association to plasma membrane scaffold proteins, some nuclear hormone receptors can be anchored to the cell membrane and act independently of their nuclear function [30]. Therefore, any signaling dysregulation or unregulated production of sex steroid hormones and/or their receptors can lead to aberrant growth and development as well as the initiation and proliferation of hormone-sensitive cancers [31].

## 2. Androgen Receptors as a Potential Culprit of Sex-Related Disparities in HNSCC

Several studies have shown that the sex hormone receptors (SHRs) can be prognostic and potential therapeutic targets of the HNSCC because they promote DNA hypermutation and potentiate HPV integration [32]. As already mentioned, due to the high male–female incidence and mortality rate ratios, endocrine homeostasis disruption is one of the potential risk factors for HNSCC development together with well-established risk factors such as alcohol and tobacco consumption. Park et al. recently published a large follow-up study that compared sex differences in HNSCC prevalence in which they showed the most evident differences between the sexes in the incidence of the upper aerodigestive tract tumors, primary laryngeal, and hypopharyngeal [33]. A higher prevalence of laryngeal cancer in males might be connected to hormonal control because, during puberty, the larynx goes through different physiological changes related to sex hormone receptors [33]. Moreover, the destruction of liver function and metabolic processes (including steroidogenesis) related to heavy alcohol consumption is another connection between the steroid receptors and HNSCC prevalence [34]. Therefore, the sex hormone receptors represent a potential therapeutic target for applying hormone receptor modulators and significant predictors of disease and treatment outcomes in some HNSCC subgroups.

### 2.1. Structure, Function, and Role of Nuclear Androgen Receptor in HNSCC

The nuclear androgen receptor (nAR) together with the estrogen, progesterone, and corticosteroid receptors belong to the steroid hormone receptor subfamily of the nuclear receptor superfamily. Structural elements of domains enable affinity binding to their ligands and response elements of the target genes for direct regulation of gene transcription [35]. The *AR* gene is located on the X chromosome (Xp11-12) and contains eight exons and seven introns that encode four structural domains that are common to all SHRs [36]. The prominent variability among the SHR family members is due to the amino acid sequence differences and the size of the N-terminal domain (NTD). This domain contains the ligand-independent activation function region AF1 and acts as a transcription driver or repressor depending on post-translational modifications [37,38]. Further, located next to the NTD is a highly conserved DNA binding domain (DBD) that contains two zinc finger motifs of four cysteine residues in each of them. Zinc finger proteins recognize DNA consensus sequences and promote the binding of the nAR to the androgen response elements (AREs) in promoters and enhancers of AR-regulated genes [39]. The short region between the DBD and C-terminal domain or ligand-binding domain (LBD) is the hinge region. Like the NTD, the hinge region is poorly conserved among the nuclear steroid receptors. Except for the nuclear localization signal, which can be contained within the hinge region, this region is also a site of post-translational modifications [38]. The LBD is a conserved domain amongst the nuclear receptors located at the C-terminal end of the receptor, and its structure creates a variable hydrophobic region for ligand binding and exhibits a ligand-dependent activation function motif (AF2) [40]. Furthermore, this region interacts with transcriptional intermediary factors (TIFs), co-activators, and co-repressors upon interaction with the ligand as well as with the N-terminal domain due to the stabilization of bounded androgens [41]. Importing the receptor into the nucleus is mediated by the nuclear localization signal (NLS) located between the DBD and hinge region. In contrast, the nuclear export signal (NES), as a part of the LBD, is responsible for the export of the nAR to the cytoplasm [42].

The nuclear androgen receptor is a 110 kDa ligand-inducible protein composed of 919 amino acids that is located in the cytoplasm and has two main mechanisms of action. The binding of androgens to the nAR results in a conformational change and phosphorylation of the protein as well as translocation of the androgen/nAR complex to the nucleus in the form of a homodimer. Once dimerized, nAR binds to the AREs within target genes and modulates gene transcription with various co-regulators [43] (Figure 2). In addition to the ligand binding, an alternative or noncanonical mechanism of nAR activation mediated by the activation of secondary signaling pathways such as mitogen-activated protein kinase (MAPK), protein kinase B (Akt), and extracellular signal-regulated kinase (ERK) also exists [44]. Interestingly, Trnski et al. recently showed the noncanonical activation of nAR via the active form of Sonic hedgehog protein (SHH-N) binding directly to the nAR through its cholesterol modification in androgen-independent LNCaP prostate cancer cells [45]. This interaction is one of the possible mechanisms of nAR activation in tumors.

AREs contain a pair of conserved sequences of 5’-GGTACAnnnTGTTCT-3’ that often are arranged as inverted repeats separated by a three-nucleotide spacer (5’-CGG-3’) [46,47]. According to the JASPAR 2022 database (https://jaspar.uio.no/matrix/MA0007.2 accessed on 21 December 2022), there are 11,206 AREs in the human genome (build GRCh37/hg19) [48]. However, so far fewer than 2000 human genes have been characterized as androgen-responsive (those whose transcription could be regulated either positively or negatively by nAR) [49]. The development of next-generation sequencing techniques—especially whole transcriptome sequencing (RNA-Seq) and chromatin immunoprecipitation followed by sequencing (ChIP-seq)—has enabled the examination of nAR-mediated transcriptional regulation at the genomic scale. Despite some differences in sets of discovered genes, which was due to the use of different sequencing platforms and experimental conditions, a core set of androgen-responsive genes has been repeatedly identified [50]. According to the Molecular Signatures Database (MSigDB), there are 101 hallmark androgen-responsive genes (MsigDB systematic name M5908) (Figure 2) [51]. These genes belong to several general gene families such as those that encode proteins involved in the secretory pathway, polyamine synthesis and lipogenesis, transcription, splicing, ribosomal biogenesis, mitogenesis, bioenergetics, and redox processes [52]. Dysregulation of those pathways is related to cancer etiopathology, so in addition to the well-known role of nAR in prostate cancer [53], nAR’s connection with breast [54], bladder [55], hepatocellular [56], ovarian [57], endometrial [58], uterine [59], and many other tumor types [60] is also known.

Currently, there is limited evidence of the significance of nAR expression in different subtypes of head and neck cancers. For example, immunohistochemical staining determined the nAR expression in 64% of laryngeal carcinoma cases; the expression was higher in well-differentiated cases and lower in poorly differentiated cases and those with lymphatic invasion [61]. Similarly, Fei et al. determined the significantly upregulated expression of nAR mRNA in laryngeal squamous cell carcinoma compared to the healthy tissue; in addition, AR-positive (AR+) nuclear immunostaining was observed in 77.1% of the samples [62]. Therefore, AR+ cytoplasmic staining in the neoplastic oral squamous cell carcinoma (OSCC) epithelium greater than 20% was significantly correlated with the increased risk of OSCC progression [63] and frequency of VEGF-positive lymphocytes and Ki67 in metastases [64]. In addition, in nonsquamous salivary duct carcinoma (SDC), nAR is one of the potential diagnostic immunohistochemical markers [65]. Furthermore, several studies have reported significantly increased nAR expression as well as the expression of androgen-responsive genes and an enhanced growth rate of the SCC-4, SCC-25, OECM-1 and SAS OSCC cell lines after treatment with dihydrotestosterone (DHT) [66]. Likewise, the nAR knockdown resulted in a 75% lower growth rate, significant apoptotic cell death, and inhibition of tumorigenicity in the above-mentioned cells [66]. Similarly, the DHT stimulation of nAR promoted cell migration and aggressiveness by increasing phosphorylation of the epidermal growth factor receptor (EGFR) and Akt in AR+ OSCC tumors. At the same time, this effect was not observed in an AR-negative (AR-) SCC-25 cell line, which suggested that nAR promotes the EGFR signaling pathway in head and neck tumors, therefore making it a perfect target for targeted therapy in this tumor type [67]. On the other hand, Collela et al. showed contradictory results; in fact, they found a lower expression level of nAR in OSCC than in the control samples [68]. Furthermore, the alterations in the [CAG]_n_ repeats of the *AR* gene are also associated with poor outcomes in male patients with oral cavity or oropharyngeal cancers; shorter repeats (≤20) were correlated with a more aggressive tumor subset [69]. Taken together, in HNSCC patients, nuclear AR+ staining generally prevails and is associated with well- and moderately differentiated tumors without lymphatic invasion, while cytoplasmic AR+ staining is correlated with metastases. Table 1 summarizes the current knowledge about the role of the nuclear and currently known membrane ARs in HNSCC.

### 2.2. Structure, Function, and Role of Membrane Androgen Receptors in HNSCC

In contrast to the nuclear receptors, membrane steroid receptors mediate the rapid nongenomic effects of steroids through the activation of secondary messengers, different signaling pathways, and calcium ion flux. Immediate androgen actions have been described in various cell and tissue types. Research on membrane androgen receptors (mARs) has accelerated after rapid intracellular signaling pathway activation in AR- cells was observed [76]. So far, five multifunctional proteins have been described as plasma mARs: Ca^2+^ ion channels TRPM8 and CaV1.2, G-protein-coupled receptors OXER1 and GPRC6A, and zinc transporter ZIP9 (Figure 2).

#### 2.2.1. Transient Receptor Potential Cation Channel Subfamily M Member 8 (TRPM8)

TRPM8 is the first of two calcium ion channels and androgen-related receptors at once located at the plasma membrane and endoplasmic reticulum of the cells. Due to its expression in the peripheral nervous system, in pain- and temperature-sensing neurons, respectively, it was shown that TRPM8 responded to cold physical stimulations, menthol, and icilin [77]. In addition, testosterone was shown to increase intracellular Ca^2+^ influx and activate the TRPM8 channel in PC-3 prostate cancer cells, dorsal root ganglion and hippocampal neurons, and the human embryonic kidney cell line HEK293 with almost the same affinity and specificity as for the nAR [78]. On the other hand, in the HEK293 cell line, which stably expresses recombinant TRPM8, testosterone did not cause significant Ca^2+^ oscillations. However, even that effect was partially reversed using *AR* siRNA or the nAR antagonist hydroxyflutamide, thereby indicating the importance of nAR in the absence of TRPM8 [79]. Furthermore, sustained exposure of the TRMP8 to testosterone caused channel desensitization—especially in men—due to higher circulating levels of testosterone [79]. TRPM8 is a potential early-stage prostate cancer biomarker because it is significantly upregulated in early stages and decreased in advanced stages of prostate tumors [80]. Likewise, TRPM8 is also a predicted diagnostic indicator of breast cancer, which promotes metastatic potential by activating the Akt/GSK-3β signaling pathway and thus regulates the epithelial–mesenchymal transition [81] as well as cellular autophagy via activation of AMP-activated protein kinase (AMPK) and Unc-51 like autophagy activating kinase 1 (ULK1) [82].

Studies have shown both intracellular and plasma membrane expression of the TRPM8 channel in HNSCC cell lines, contrary to exclusively membranous localization of other mARs. Interestingly, since cancer cells secrete matrix metalloproteinases, which degrade the basement membrane and the extracellular matrix and thus facilitate the cell migration and invasion ability, this mechanism was shown to potentially be associated with an elevated intracellular Ca^2+^ level in the androgen-dependent LNCaP prostate cancer cell line [83]. Similarly, the menthol-induced activity of the TRPM8 enhanced the migration and invasion potential of the HSC-3 and HSC-4 OSCC cell lines via an increase in the activation and gelatinase activity of matrix metallopeptidase 9 (MMP-9). On the other hand, the TRPM8 antagonist RQ-00203078 suppressed this activity, which suggested that stimulation of TRPM8 positively regulates MMP-9 [70]. Recent bioinformatical analysis of the HNSCC dataset from The Cancer Genome Atlas (TCGA-HNSC) indicated TRPM8 as one of the risk factors in HNSCC evolvement [71], but its main role in this type of tumorigenesis is still unknown.

#### 2.2.2. Voltage-Dependent L-Type Calcium Channel Subunit Alpha-1C (CaV1.2)

CaV1.2 is a protein encoded by the *CACNA1C* gene whose primary role is pore formation for Ca^2+^ ion transport into the cell. It is a member of the L-type voltage-gated Ca^2+^ channel family [84]. The nomenclature is based on the type of response to voltage, meaning this type of channel has ‘long-lasting’ activity, and its activation requires a strong membrane depolarization [85]. Furthermore, the protein consists of three subunits (CaVα1, CaVα2δ, and CaVβ), and four CaVα1 subunits set up the ion-conducting channel. Alternative splice variants of the *CACNA1C* gene are associated with various abnormalities such as Timothy and Brugada syndromes and cancer progression [85,86,87]. Increased influx of the Ca^2+^ ions through the plasma membrane is related to the activation of Fos and c-Jun transcription factors, cAMP response element-binding (CREB) protein, and the nuclear factor of activated T cells (NFAT) [88]. Likewise, Ca^2+^ is indispensable in coordinating transitions between G1/S phases of the cell cycle via regulation of D-cyclin expression. All the above changes are implicated in breast cancer cell proliferation, migration, and invasion [89].

When comparing the gene expression profiles in OSCC-derived cell lines with normal oral tissues using a microarray analysis, *CACNA1C* was one of the detected genes with a confirmed significantly decreased expression in the HSC-2, HSC-3, Ca9-22, H-1, Sa3, and OK-92 OSCC cell lines compared to human normal oral keratinocytes [72]. In addition, another study compared genes dysregulated in the TCGA-HNSC dataset with the early molecular alterations in the oral cavity and esophagus induced by (S)-N’-nitrosonornicotine, a potent tobacco carcinogen, in rats; except for the genes involved in immune regulation and inflammation, most of the dysregulated genes in rats were tumorigenesis-associated and overlapped with altered genes in esophageal and head and neck tumors. This included *CACNA1C*, which was amplificated and/or upregulated in 10% of cases [73].

#### 2.2.3. Oxoeicosanoid Receptor 1 (OXER1)

OXER1 is a G-protein-coupled receptor (GPCR) that is also known as G-protein-coupled receptor 170 (GPR170), 5-oxo-ETE G-protein-coupled receptor, or TG1019. The receptor can be activated by 5-oxoeicosatretraenoic acid (5-oxo-ETE) or some other product of the arachidonic acid metabolites and by 5-lipoxygenase (5-LOX) and peroxidase [90]. High expression of OXER1 in lung, liver, spleen, kidney, prostate, and breast cancer cells as well as in inflammatory cells (eosinophils, lymphocytes, monocytes, and neutrophils) and its activation by endogenous ligand 5-oxo-ETE mediated the intracellular actions in cell proliferation and survival, inflammatory responses, and steroidogenesis stimulation [91]. On the one hand, receptor activation led to the inhibition of cAMP production mediated by the G_αi_ subunit. In contrast, the calcium mobilization, chemotactic response, and activation of other molecular pathways were mediated by a stable dimeric complex G_βγ_ [90]. Most of these roles are related to the immune system, but OXER1 is also essential in promoting survival, apoptosis inhibition, and infiltration of inflammatory cells in prostate and breast cancers. Recently, it was reported that testosterone is an OXER1 antagonist that mediated the effects of 5-oxo-ETE on PI3K/Akt, FAK, and p38α activation [92]. The inhibition of the above signaling pathways (such as blocking 5-LOX) reduces cancer cell proliferation, migration, invasion, and apoptosis induction.

So far, no direct studies have investigated the role of OXER1 in HNSCC, but upregulation of the *OXER1* gene was reported in HPV+ oropharyngeal and oral cancers [74]. As stated above, HPV infection has an essential role in HNSCC progression. Thus, OXER1 is potentially connected with HNSCC tumorigenesis and could present a novel drug target. Interestingly, López-Ozuna et al. identified the upregulated expression of *OXER1* in 2N and 11N human normal oral epithelial cell lines exposed to water-pipe smoking (WPS) solution, thereby implying the potential risk of WPS consumption for HNSCC development [75].

#### 2.2.4. G-Protein-Coupled Receptor Class C Group 6 Member A (GPRC6A)

GPRC6A is the second GPCR mAR. It is activated by a diverse range of ligands such as essential L-α-amino acids (L-arginine, L-lysine, and L-ornithine), cations (calcium, zinc, and magnesium), osteocalcin [93], and steroid hormones [94]. The expression of GPRC6A in hepatocytes, skeletal muscle myocytes, and adipocytes is related to its involvement in different physiological and pathological functions such as glucose and fat metabolism, inflammatory responses, steroid production and insulin secretion from β-cells, and tumorigenesis [95]. Furthermore, the role of GPRC6A in the male reproductive system was recently discovered. Oury et al. showed expression of GPRC6A in Leydig cells, where it regulated spermatogenesis and testosterone biosynthesis in male mice by acting as an osteocalcin receptor [96]. In addition, they demonstrated a reduction in circulating testosterone levels, a decreased sperm number, and a reduced size of the testis in GPRC6A_Leydig_ KO mice. Likewise, in an AR- and GPRC6A-negative HEK293 cell line, testosterone and synthetic androgen R1881 could stimulate ERK activity in a dose-dependent manner after transfection with human GPRC6A. On the other hand, ERK activation was blocked using a G-protein inhibitor, which confirmed the hypothesis of androgen-mediated GPRC6A activation [94]. Additionally, it recently was shown that due to the similar structure of the undercarboxylated form of osteocalcin and the sex hormone-binding protein (SHBG), the latter—in an unliganded state—binds to the extracellular domain of GPRC6A and activates the receptor [97]. Computational modeling also predicted several testosterone binding sites on the GPRC6A receptor, although the specificity of its binding and GPRC6A receptor activation in vivo is still unclear [98]. Several studies confirmed a connection between *GPRC6A* upregulation and prostate cancer cell line proliferation [99,100,101], while there is still no evidence for the potential role of the GPRC6A receptor in HNSCC tumorigenesis.

#### 2.2.5. Zinc Transporter Member 9 (ZIP9)

ZIP9 is a member of the zinc transporter Zrt- and Irt-like protein (ZIP) family encoded by the *SLC39A9* gene in humans. The central role of ZIP transporters is the regulation of zinc homeostasis by increasing the intracellular zinc concentration and therefore maintaining the normal physiological function of the organism. Compared to the other ZIP family members that contain eight transmembrane domains and an extracellular C-terminus, ZIP9 is the only member with seven transmembrane domains and an intracellular C-terminal domain, which enables signalization via G proteins [102]. In addition to regulating cellular functions associated with zinc transport, ZIP9 was first identified as a mAR in Atlantic croaker fish ovarian follicle cells [103] and then immediately afterwards in the triple-negative human breast cancer cell lines MDA-MB-468 and MDA-MB-231 as well as in the PC-3 prostate cancer cell line, all of which were stably transfected with ZIP9 [104]. Treatment with testosterone in those cell lines led to G-protein activation, thereby increasing the cytosolic zinc concentrations and apoptosis induction through the upregulation of the proapoptotic genes *BAX*, *TP53,* and *MAPK8*. On the other hand, testosterone-induced increased expression of the proapoptotic factors was blocked by using a MAPK inhibitor (PD98059) or a zinc chelator, which suggested that the apoptotic response was mediated by MAPK activation and zinc influx [104]. Furthermore, when comparing the binding affinity and activation of nAR and ZIP9 via testosterone, DHT, and androstenedione, ZIP9 showed a higher affinity for testosterone regarding DHT or androstenedione [102]. Except for its role in apoptosis induction, an activated form of the ZIP9 receptor affected cell migration in prostate cancer cells [105]. Due to its broad expression in most tissues and cell types related to zinc homeostasis, ZIP9 is involved in cell proliferation, growth, and apoptosis and represents a potential drug target in different types of carcinomas. However, the connection between ZIP9 and HNSCC tumorigenesis is still unexplored.

## 3. Current Therapeutic Approaches to the Treatment of HNSCC

Since the primary treatment for most patients with HNSCC is surgery and radiotherapy, research is focused on developing effective targeted therapies and the identification of new biomarkers to reduce mortality and improve the treatment outcome as well as the life quality of the patients. The biggest challenge in this, as is the case with many other tumors, is a weak response to therapy and the development of therapy resistance due to the heterogeneous nature and aggressiveness of HNSCC. Likewise, the roles of both the tumor microenvironment and the oral microbiome in HNSCC pathogenesis are increasingly being investigated.

### 3.1. Immunotherapy Strategies for Head and Neck Cancer

Interestingly, about 70% of HPV-negative HNSCCs cases have the loss-of-function (LoF) mutations in tumor-suppressor p53 while 30% of patients have them in the mammalian target of rapamycin (mTOR) and around 15% have them in the EGFR and NOTCH oncogenes. In comparison, the EGFR is overexpressed in more than 80% of HNSCCs [10]. Therefore, to date, the FDA has fully approved three targeted immunotherapies for HNSCC treatment: the anti-EGFR monoclonal antibody cetuximab as well as pembrolizumab and nivolumab, which are programmed cell death protein 1 (PD-1) inhibitors [13]. Cetuximab mediates in an oncogenic signal blocking and killing tumor cells by activating antibody-dependent cell-mediated cytotoxicity (ADCC) via natural killer (NK) cells and monocytes [64]. It was used for a decade as the first line of HNSCC treatment together with other platinum-based drugs such as cisplatin and 5-fluorouracil (5-FU). In contrast, the latter two antibodies block the interaction of PD-1 with its programmed death-ligand 1 (PD-L1) and are used to treat recurrent or metastatic (R/M) HNSCC [106]. Since the KEYNOTE-048 trial was published in 2019, pembrolizumab has been approved as a first-line treatment for PD-L1-positive R/M HNSCC and in combination with chemotherapy for patients with any PD-L1 status [107]. However, the efficacy of targeted therapies is reduced due to the development of drug resistance. For instance, EGFR shares common downstream pathways (the RAS-RAF-MAPK and PI3K-AKT-mTOR) with alternative receptor tyrosine kinases (RTK) such as hepatocyte growth factor receptor (c-MET), human epidermal growth factor receptor 3 (HER3), and AXL receptor tyrosine kinase, so any mutations or aberrant signaling activity that lead to epidermal growth factor (EGF)-independent signal transduction can cause resistance development [108].

### 3.2. Applicability of Antiandrogen Therapy in HNSCC

On the other hand, given the evidence of elevated SHR expression in HNSCC, nAR also represents a potential target for antiandrogen therapy in these tumors (Figure 2)—notably in SDC. SDC is a very rare and aggressive subtype of salivary gland carcinoma (1–3%) with a 5-year survival rate below 20% [109]. Interestingly, nAR is expressed in 70–98% of SDC cases [110,111] and significantly more often in men than in women [112]. Therefore, based on nAR expression, in recent years the first-line treatment for AR+ SDC is androgen deprivation therapy (ADT) using the first-generation of nonsteroidal antiandrogens such as bicalutamide [113] and flutamide as well as abiraterone acetate [114], an inhibitor of androgen synthesis [115]. Several studies have shown the benefits of ADT use in AR+ SDC in terms of a higher response rate and a better prognosis and clinical outcome [109,116,117,118]. Locati et al. reported a 64.7% overall response rate and three complete clinical remissions in a cohort of 17 AR+ SDC patients who received ADT [119]. Furthermore, ADT proved less toxic but equally effective in both R/M and unresectable locally advanced SDC compared with conventional chemotherapy [120]. First-line ADT therapy also demonstrated a higher response rate than the first-line chemotherapy treatment in SDC patients [109]. Nonetheless, combining chemotherapy or radiotherapy with ADT seems to be the most effective for AR+ HNSCC treatment because nAR expression positively correlates with overall and metastases-free survival [121]. Interestingly, an in vitro study also showed that flutamide treatment of the HSY salivary gland cancer cell line could block the effects of DHT, which has been shown to increase cell proliferation, migration, and invasion [114].

Since the vast majority of SDC tumors express nAR, treatment with nonsteroidal antiandrogens is often used in combination with GnRH agonists such as triptorelin and goserelin and luteinizing hormone-releasing hormone (LHRH) agonists (leuprorelin acetate) [122]. This type of combined androgen blockade (CAB) has an established role in prostate cancer treatment, and therefore translating these combined treatment strategies to SDC was studied in a phase II study that suggested the equivalent efficacy and lower toxicity of CAB for SDC patients compared with chemotherapy [120]. Currently, in the treatment of prostate cancer, new-generation nonsteroidal antiandrogens such as abiraterone acetate, enzalutamide, apalutamide, and darolutamide also are used [123]. Enzalutamide monotherapy treatment was used for AR+ SDC patients in a phase II trial in which only 15% of patients (7/46) had a partial response and 2/46 patients maintained the response until 8 weeks [124]. On the other hand, in the first published report of SDC responding to abiraterone, a 10-month progression-free survival was described in one patient [125]. Furthermore, 19 AR+ SDC patients were treated with abiraterone plus prednisone and an LHRH agonist in another phase II trial. The results showed that this combined therapy represented an active and safe second-line treatment option for AR+ SDC with a 12-month overall survival rate of 74.5% [126]. Likewise, the first prospective trials that evaluated CAB with apalutamide and gorselin as a GnRH agonist (NCT04325828) as well as with darolutamide and gorselin for AR+ SDC treatment are currently ongoing (NCT05694819). Although SDC can be classified as an androgen-dependent tumor, the growth of which can be controlled using ADT, the effectiveness and molecular mechanisms underlying the application of ADT in this type of tumor have yet to be investigated.

## 4. Conclusions

Disparities in HNSCC incidence and mortality rates between males and females are more than evident, and this could not be associated only with the different lifestyles. Although present levels of evidence are still quite limited and scarce, potential toxic effects of male sex hormones in HNSCC development and progression either alone or in combination with other well-known risk factors could be discovered by studying the role of ARs in the etiopathology of HNSCC. At present, nAR is the most studied AR in HNSCC. However, using the expression status of nAR as a prognostic or predictive biomarker for HNSCC is still far from a routine clinical practice. The only exception is AR+ salivary duct carcinomas, for which several clinical trials have shown an advantage in using the new generation of nonsteroidal antiandrogens in terms of a higher response rate and a better prognosis and clinical outcome. In that light, the substitution of traditional ADT with new-generation drugs or even using CAB might provide a better quality of life for all patients with AR+ HNSCC.

## Figures and Tables

**Figure 1 ijms-24-03766-f001:**
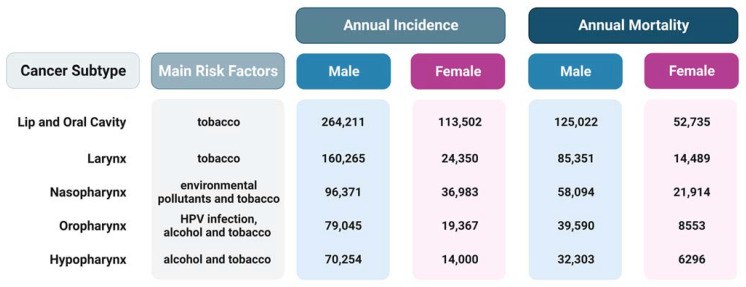
The primary risk factors and annual incidence and mortality rates in both males and females for different subtypes of head and neck squamous cell carcinoma. The data are from Global Cancer Statistics 2020 [1]. Created with BioRender.com.

**Figure 2 ijms-24-03766-f002:**
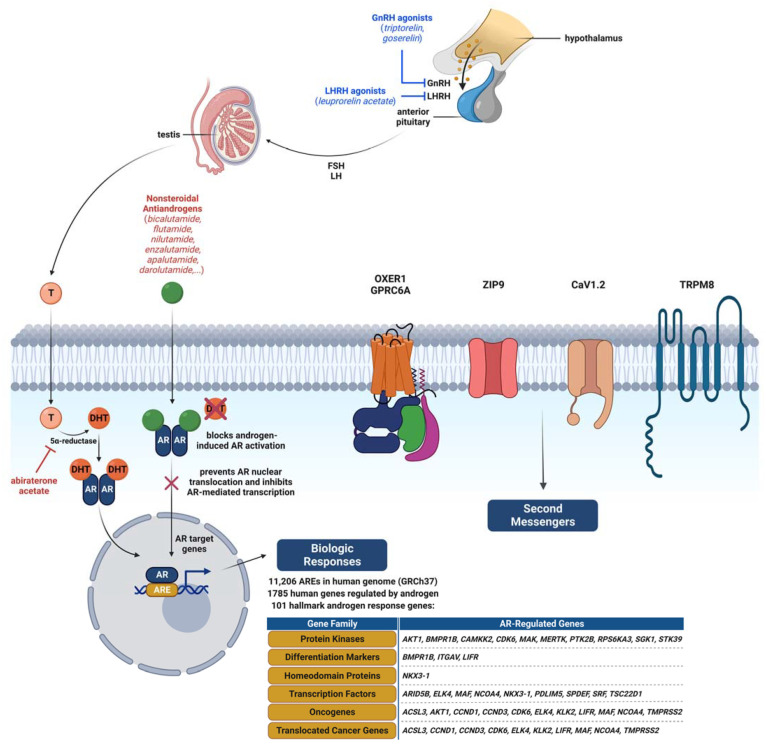
Nuclear and membrane androgen receptors and different types of antiandrogen therapy. The nuclear form of the androgen receptor (AR) is activated by binding of dihydrotestosterone (DHT), the 5α-reduced metabolite of testosterone (T), in the cytoplasm and then translocation into the nucleus. There, it serves as a transcription factor of many target genes that contain the androgen response element (ARE) in their promoters and enhancers. Many AR-regulated genes are related to carcinogenesis. Nonsteroidal antiandrogens block androgen-induced activation of AR and prevent its nuclear translocation and thus inhibit AR-mediated transcription in both the hypothalamus and target tissues. Steroidal antiandrogens, like abiraterone acetate, block enzymes involved in androgen biosynthesis, thereby stimulating the negative feedback loop in the hypothalamus, which results in lowering the plasma T concentration. Antiandrogens are often used in combination with a gonadotropin-releasing hormone (GnRH) agonist or luteinizing hormone-releasing hormone (LHRH). The currently known membrane androgen receptors are a group of five proteins: Ca^2+^ ion channels TRPM8 and CaV1.2, G-protein-coupled receptors OXER1 and GPRC6A, and zinc transporter ZIP9, which activate different signaling pathways through the second messengers. FSH—follicle-stimulating hormone; LH—luteinizing hormone. Created with BioRender.com.

**Table 1 ijms-24-03766-t001:** Currently known androgen receptors and their significance in head and neck squamous cell carcinoma.

Type of Androgen Receptor	Receptor	Receptor Function	Significance in HNSCC	Reference
Nuclear	AR	Transcription factor	Expressed in 64% of LSCC cases; higher in well-differentiated, lower in poorly differentiated and those with lymphatic invasion	[61]
			*AR* mRNA upregulated in LSCC; AR+ nuclear staining observed in 77.1% of samples	[62]
			AR+ cytoplasmic staining in OSCC epithelium greater than 20% correlated with metastases as well as the frequency of VEGF positive lymphocytes and Ki67 expression in metastases	[63,64]
			DHT treatment increased proliferation, migration, and invasion of OSCC cells by increasing EGFR phosphorylation, which was reversed by AR knockdown or observed only in AR+ cells	[66,67]
			Lower expression in OSCC samples	[68]
			In male oral cavity or oropharyngeal cancer patients, a lower number of *AR* gene [CAG]_n_ repeats (≤20) correlated with shorter disease-free survival and more frequent recurrence or metastasis	[69]
Membrane	TRPM8	Calcium ion channel	Menthol-induced Ca^2+^ influx through TRPM8 enhanced migration and invasion of OSCC cells via gelatinase activity of MMP-9	[70]
			Higher *TRPM8* mRNA expression was a negative prognostic biomarker for HNSCC	[71]
	CaV1.2	Calcium ion channel	Decreased *CACNA1C* mRNA expression observed in OSCC cells; *CACNA1C* amplification and/or mRNA upregulation observed in 10% of HNSCC cases	[72,73]
	OXER1	G-protein-coupled receptor	Upregulated *OXER1* mRNA expression observed in HPV+ oropharyngeal and oral cancers	[74]
			Upregulated *OXER1* mRNA expression observed in human normal oral epithelial cells exposed to water-pipe smoking	[75]
	GPRC6A	G-protein-coupled receptor	Unknown	/
	ZIP9	Zinc transporter	Unknown	/

AR+—androgen receptor-positive; DHT—dihydrotestosterone; HNSCC—head and neck squamous cell carcinoma; HPV+—human papillomavirus-positive; LSCC—laryngeal squamous cell carcinoma; OSCC—oral squamous cell carcinoma.

## Data Availability

Not applicable.

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
