# Peer review of "‘Toxic Masculinity’: What Is Known about the Role of Androgen Receptors in Head and Neck Squamous Cell Carcinoma"

_ijms, 2023, doi:10.3390/ijms24043766_

Round 1
Reviewer 1 Report
This review article discussed toxic masculinity in view of what is known about the role of androgen receptors in head and neck squamous cell carcinoma. There are some concerns in this manuscript as follows:
1. Title: The sentence “Toxic masculinity” should not be mentioned in the title because this review is a general overview regarding the role of androgen receptors in head and neck squamous cell carcinoma and methods of their management. The manuscript didn’t focus on toxic masculinity as definition, etiology, and manifestations. Please, revise.
2. I think that the “Introduction” shouldn’t have subheadings. Please, revise.
3. The novel points in this review article should be clarified because there are previous reviews that discussed similar issues; e.g. https://pubmed.ncbi.nlm.nih.gov/31299692/
4. Page 2 Lines 55-58: The sentence “Accordingly, new targeted therapies and predictive, prognostic, and diagnostic biomarkers for early clinical detection of HNSCC are necessary. Various new biomarkers were evaluated in clinical trials in the last decade, including hormonal receptors” has no references. Please, add a reference.
5. Page 3 Line 117: The word “Receptor” should be replaced with “Receptors”. Please, revise
6. Page 3 Lines 122-127: The sentence “AR gene is located on the X chromosome (Xp11-12) and contains eight exons and seven introns encoding four structural domains that are common for all SHRs. The prominent variability among the SHR family members is due to the amino acid sequence differences and the size of the N-terminal domain (NTD). This domain contains ligand-inde-pendent activation function region AF1 and acts as a transcription driver or repressor, depending on post-translational modifications” has no references. Please, add references.
7. Page 4 Lines 134-138: The sentence “Except for the nuclear localization signal, which can be contained within the hinge region, this region is also a site of post-translational modifications. The LBD is a conserved do-main amongst the nuclear receptors located at the C-terminal end of the receptor, and its structure creates a variable hydrophobic region for ligand binding and exhibits a ligand-dependent activation function motif (AF-2)” has no references. Please, add a reference.
8. Page 4 Lines 141-144: The sentence “Importing the receptor into the nucleus is mediated by the nuclear localization signal (NLS) located between the DBD and hinge region. In contrast, the nuclear export signal (NES), as s part of the ligand-binding domain, is responsible for the export of the nAR to the cytoplasm.” had no references. Please, add.
9. It is recommended to write the subheadings of Membrane Androgen Receptors in pages 6-8 in detailed form and not as abbreviations; i.e. TRPM8, CAC1C, etc….
10. In page 7, the first paragraph has only two references (60, 61). Please, add more references to this paragraph.
11. Page 9: The subheading “Current Therapeutic Approaches“ should be changed to “Current Immunotherapeutic Approaches” because the paragraph mentioned only the role of immunotherapy in HNSCC.
12. Page 9: More details about the applicability of anti-androgen therapy in HNSCC regarding their efficacy and potential adverse effects should be added. Also, the role of other members of second and third generation antiandrogens such as Enzalutamide and Apalutamide and the combination therapeutic strategies should be discussed.
13. A figure summarizing the mechanism of action of the different Anti-androgen therapies should be added.
14. I think that the conclusion should be summarized to focus on the possible clinical implications of the data obtained from the present review.
15. The manuscript should be thoroughly checked regarding the grammatical and typing errors.
16. The meaning of the abbreviations should be clearly defined at their first mention.
Author Response
This review article discussed toxic masculinity in view of what is known about the role of androgen receptors in head and neck squamous cell carcinoma. There are some concerns in this manuscript as follows:
We thank the reviewer for carefully and thoroughly reading our manuscript and proposing valuable suggestions to further improve our manuscript.
- Title: The sentence “Toxic masculinity” should not be mentioned in the title because this review is a general overview regarding the role of androgen receptors in head and neck squamous cell carcinoma and methods of their management. The manuscript didn’t focus on toxic masculinity as definition, etiology, and manifestations. Please, revise.
We are well aware of actual meaning of the phrase ‘toxic masculinity’ which is used in social sciences and psychology, and that it has nothing with the topic of our review manuscript. However, we think that it is a perfect metaphor for toxic (carcinogenic) roles of masculine (male) sex hormones in HNSCC! Therefore, by using this phrase in the title, we hope this paper would gain more attention. On the other hand, to avoid any potential misunderstandings, the phrase “toxic masculinity” will be put under quotations.
- I think that the “Introduction” shouldn’t have subheadings. Please, revise.
We have checked, there are several recent review papers in IJMS which have subheadings in the ‘Introduction’ chapter (e.g., https://doi.org/10.3390/ijms24021782 and https://doi.org/10.3390/ijms24021783), so this should be fine.
- The novel points in this review article should be clarified because there are previous reviews that discussed similar issues; e.g. https://pubmed.ncbi.nlm.nih.gov/31299692/.
As explained in an answer to your first concern, our manuscript has nothing with certain male behaviors, so suggested reference is out of the scope of our manuscript.
- Page 2 Lines 55-58: The sentence “Accordingly, new targeted therapies and predictive, prognostic, and diagnostic biomarkers for early clinical detection of HNSCC are necessary. Various new biomarkers were evaluated in clinical trials in the last decade, including hormonal receptors” has no references. Please, add a reference.
References 12-15 have been added.
- Page 3 Line 117: The word “Receptor” should be replaced with “Receptors”. Please, revise
Thank you for your comment, but since there is only one type of nuclear androgen receptor, we assume that the word ˝Receptor˝ is correct.
- Page 3 Lines 122-127: The sentence “AR gene is located on the X chromosome (Xp11-12) and contains eight exons and seven introns encoding four structural domains that are common for all SHRs. The prominent variability among the SHR family members is due to the amino acid sequence differences and the size of the N-terminal domain (NTD). This domain contains ligand-inde-pendent activation function region AF1 and acts as a transcription driver or repressor, depending on post-translational modifications” has no references. Please, add references.
References 36-38 have been added.
- Page 4 Lines 134-138: The sentence “Except for the nuclear localization signal, which can be contained within the hinge region, this region is also a site of post-translational modifications. The LBD is a conserved do-main amongst the nuclear receptors located at the C-terminal end of the receptor, and its structure creates a variable hydrophobic region for ligand binding and exhibits a ligand-dependent activation function motif (AF-2)” has no references. Please, add a reference.
References 38 and 40 have been added.
- Page 4 Lines 141-144: The sentence “Importing the receptor into the nucleus is mediated by the nuclear localization signal (NLS) located between the DBD and hinge region. In contrast, the nuclear export signal (NES), as s part of the ligand-binding domain, is responsible for the export of the nAR to the cytoplasm.” had no references. Please, add.
Reference 42 has been added.
- It is recommended to write the subheadings of Membrane Androgen Receptors in pages 6-8 in detailed form and not as abbreviations; i.e. TRPM8, CAC1C, etc….
Thank you for your comment and suggestion. We have changed the titles of subheadings from abbreviations to detailed forms.
- In page 7, the first paragraph has only two references (60, 61). Please, add more references to this paragraph.
References 84 and 86-88 have been added.
- Page 9: The subheading “Current Therapeutic Approaches“ should be changed to “Current Immunotherapeutic Approaches” because the paragraph mentioned only the role of immunotherapy in HNSCC.
Thank you for your suggestion. We didn't change the name of the subheading "Current Therapeutic Approaches" since we introduced a new subheading "3.1. Immunotherapy strategies for head and neck cancer", which now contains a text on the use of immunotherapy in the treatment of HNSCC, and in the subsection "3.2. Applicability of Anti-Androgen Therapy in HNSCC" more details have been added about this therapy type.
- Page 9: More details about the applicability of anti-androgen therapy in HNSCC regarding their efficacy and potential adverse effects should be added. Also, the role of other members of second and third generation antiandrogens such as Enzalutamide and Apalutamide and the combination therapeutic strategies should be discussed.
Thank you for your suggestion about the ADT in HNSCC. As suggested, we expanded the subchapter based on androgen deprivation therapy (˝3.2. Applicability of Anti-Androgen Therapy in HNSCC˝), where the most recent knowledge about the new generation of nonsteroidal antiandrogens and combinations with GnRH and LHRH agonists in SDC treatment is presented (Page 11, Lines 504-552).
- A figure summarizing the mechanism of action of the different Anti-androgen therapies should be added.
The mechanism of different anti-androgen therapies, steroidal and nonsteroidal antiandrogens, as well as GnRH and LHRH agonists, has been summarized as part of Figure 2.
- I think that the conclusion should be summarized to focus on the possible clinical implications of the data obtained from the present review.
As suggested, the new conclusion has been added.
- The manuscript should be thoroughly checked regarding the grammatical and typing errors.
The text has been thoroughly inspected and all observed grammatical and typing errors were corrected.
- The meaning of the abbreviations should be clearly defined at their first mention.
All of the text's abbreviations were checked and are now defined at their first mention.
Reviewer 2 Report
Authors have provided an adequate review of the subject, illustrated with good quality tables and figures. Following this review, a real life case study is needed to try to verify all that has been presented in.
Author Response
Comments and Suggestions for Authors
Authors have provided an adequate review of the subject, illustrated with good quality tables and figures. Following this review, a real life case study is needed to try to verify all that has been presented in.
We thank the reviewer for carefully and thoroughly reading our manuscript. We have already started in vitro study on the role of sex hormone receptors in HNSCC, and data presented in this manuscript is of great help for us.
Reviewer 3 Report
The manuscript is very interesting. However, it needs more depth.
1.Although Table 1 shows AR as transcription factor, it does not show what is targeted genes, whose mRNA is upregulated by AR as transcription factor.
2. The mechanism how menthol-induced TRPM8 activate MMP-9 is not shown.
Therefore, I think that authors should revise Figure2 and Table1 etc.
Author Response
Comments and Suggestions for Authors
The manuscript is very interesting. However, it needs more depth.
We thank the reviewer for carefully and thoroughly reading our manuscript and proposing valuable suggestions to further improve our manuscript.
- Although Table 1 shows AR as transcription factor, it does not show what is targeted genes, whose mRNA is upregulated by AR as transcription factor.
On Page 6, Lines 198-217, we summarized the current knowledge about the androgen-responsive genes which encode proteins involved in the secretory pathway, polyamine synthesis and lipogenesis, transcription, splicing, ribosomal biogenesis, mitogenesis, bioenergetics and redox processes. Furthermore, the main gene families with examples of genes belonging to a particular group are summarized in Figure 2.
- The mechanism how menthol-induced TRPM8 activate MMP-9 is not shown.
Thank you for your comment. The activation and positive regulation of MMP-9 via the menthol-induced TRPM8 stimulation is now explained in detail on Page 8, Lines 327-335 as well as in Table 1.
Therefore, I think that authors should revise Figure2 and Table1 etc.
Figure 2 and Table 1 have been revised accordingly.
Reviewer 4 Report
Dear editor:
The authors conducted a comprehensive review regarding the potential role of androgen receptors on head and neck squamous cell carcinoma. They concluded that nAR has a more significant potential role and can be a treatment target for head and neck squamous cell carcinoma. However, the role of mAR needs further examinations. We believe that androgen receptor will be a novel target for head and neck squamous cell carcinoma and it is worthy for comprehensive review. Nonetheless, some issues should be elucidated more detailly.
1. The author review the role of androgen receptors in head and neck squamous cell carcinoma. However, as we known, androgen receptors also plays an important role in salivary gland carcinoma. Given salivary gland carcinoma is one part of head and neck cancers, we suggest to add the review of the role of androgen receptors in salivary gland cancer, in terms of incidence, prognosis and treatment.
2. The authors could summarize the clinical or pathologic characteristic of AR positive head and neck squamous cell carcinoma for the better understanding who are the candidates.
3. The authors mentioned about androgen receptors can be divided into two categories : nAR and mAR. Given the different roles between nAR and mAR, it is important for physicians to realize which type of AR do their patients have. Thus, the authors should have some illustration regarding the methodology to detect nAR or mAR.
4. This review focus on the role of androgen receptors in head and neck squamous cell carcinoma. Thus, it is not necessary to review current treatment approaches, including EGFR targeted therapy and PD-1 immunotherapy. The authors should address more regarding the efficacy of androgen deprivation therapy in vivo and in vitro for head and neck squamous cell carcinoma.
5. The authors presented that ADT with bicalutamide, flutamide and abiraterone. How about the role of enzalutamide and GnRH agonist in AR positive head and neck squamous cell carcinoma. More cell line studies or clinical trials are warranted. Moreover, the authors suggested ADT as first line treatment for nAR positive head and neck squamous cell carcinoma. Are there any evidences regarding the treatment in patients with mAR positive head and neck squamous cell carcinoma.
Author Response
Comments and Suggestions for Authors
Dear editor:
The authors conducted a comprehensive review regarding the potential role of androgen receptors on head and neck squamous cell carcinoma. They concluded that nAR has a more significant potential role and can be a treatment target for head and neck squamous cell carcinoma. However, the role of mAR needs further examinations. We believe that androgen receptor will be a novel target for head and neck squamous cell carcinoma and it is worthy for comprehensive review. Nonetheless, some issues should be elucidated more detailly.
We thank the reviewer for carefully and thoroughly reading our manuscript and proposing valuable suggestions to further improve our manuscript.
- The author review the role of androgen receptors in head and neck squamous cell carcinoma. However, as we known, androgen receptors also plays an important role in salivary gland carcinoma. Given salivary gland carcinoma is one part of head and neck cancers, we suggest to add the review of the role of androgen receptors in salivary gland cancer, in terms of incidence, prognosis and treatment.
Salivary gland cancers (SGC) comprise about 3% of all head and neck cancers, and there are many types of SGC. The most common type of SGC is mucoepidermoid carcinoma, while HNSCC belongs to the rare SGC group. On the other hand, squamous cell skin cancer might start elsewhere and spread to the salivary glands. Since SGC represents a rare subtype of HNSCC and AR+ salivary duct carcinoma (1-3% of all SGC cases) is the primary type of cancer in which androgen deprivation therapy is used, we added only a short introduction to the SDC incidence and prognosis (Page 11, Lines 507-509) and detailed description of its treatment using androgen deprivation therapy (Page 11, Lines 504-552).
- The authors could summarize the clinical or pathologic characteristic of AR positive head and neck squamous cell carcinoma for the better understanding who are the candidates.
Known clinical and pathologic characteristics of AR+ HNSCC patients now are summarized in Table 1.
- The authors mentioned about androgen receptors can be divided into two categories : nAR and mAR. Given the different roles between nAR and mAR, it is important for physicians to realize which type of AR do their patients have. Thus, the authors should have some illustration regarding the methodology to detect nAR or mAR.
The only currently used methodology for AR detection in the clinic is immunohistochemical staining of tissue samples, which was mentioned in the text for a few times (Lines 219, 224-225, 228-229). Furthermore, only nAR staining is used as a potential diagnostic immunohistochemical marker for salivary duct carcinoma (Lines 228-229), while mAR detection is still not in use in the clinic. All other methods of detection are used only as part of in vitro and in vivo studies.
- This review focus on the role of androgen receptors in head and neck squamous cell carcinoma. Thus, it is not necessary to review current treatment approaches, including EGFR targeted therapy and PD-1 immunotherapy. The authors should address more regarding the efficacy of androgen deprivation therapy in vivo and in vitro for head and neck squamous cell carcinoma.
Thank you for your comment. Since the androgen deprivation therapy in head and neck squamous cell carcinoma is used only for salivary gland cancer treatment, we have decided to leave a review of current treatment approaches as a part of this review (new subheading ˝3.1. Immunotherapy strategies for head and neck cancer˝ added), but also to expand subchapter based on androgen deprivation therapy (˝3.2. Applicability of Anti-Androgen Therapy in HNSCC˝), as suggested.
- The authors presented that ADT with bicalutamide, flutamide and abiraterone. How about the role of enzalutamide and GnRH agonist in AR positive head and neck squamous cell carcinoma. More cell line studies or clinical trials are warranted. Moreover, the authors suggested ADT as first line treatment for nAR positive head and neck squamous cell carcinoma. Are there any evidences regarding the treatment in patients with mAR positive head and neck squamous cell carcinoma.
Further to the previous comment, in newly added on Page 11, Lines 522-552, we presented the most recent knowledge about the new generation of nonsteroidal antiandrogens, like abiraterone acetate, enzalutamide, apalutamide and darolutamide, which combined with GnRH and LHRH agonists represent a future in AR+ salivary duct cancer treatment. According to our knowledge, there is still no evidence regarding androgen deprivation therapy treatment in mAR+ HNSCC patients.